# Exploring the Potential of Sensing for Breast Cancer Detection

**Nure Alam Chowdhury** [1] , **Lulu Wang** [2,*] , **Linxia Gu** [1]  **and Mehmet Kaya** [1,*] 

1   Department of Biomedical Engineering and Science, Florida Institute of Technology,
    Melbourne, FL 32901, USA; nurealam1743phy@gmail.com (N.A.C.); gul@fit.edu (L.G.)
2   Biomedical Device Innovation Center, Shenzhen Technology University, Shenzhen 518118, China
*   Correspondence: lwang381@hotmail.com or wanglulu@sztu.edu.cn (L.W.); mkaya@fit.edu (M.K.)

**Abstract:** Breast cancer is a generalized global problem. Biomarkers are the active substances that have been considered as the signature of the existence and evolution of cancer. Early screening of different biomarkers associated with breast cancer can help doctors to design a treatment plan. However, each screening technique for breast cancer has some limitations. In most cases, a single technique can detect a single biomarker at a specific time. In this study, we address different types of biomarkers associated with breast cancer. This review article presents a detailed picture of different techniques and each technique's associated mechanism, sensitivity, limit of detection, and linear range for breast cancer detection at early stages. The limitations of existing approaches require researchers to modify and develop new methods to identify cancer biomarkers at early stages.

**Keywords:** biosensor; NMRI; microwave imaging; biomarker; breast cancer

## 1. Introduction

The abnormal growth of cells in the breast is responsible for breast cancer, which is more commonly observed in women than men [1–3]. Noninvasive cancer does not spread outside the tissue from the point at which it is generated [4]. Invasive cancer does not live in a single point; instead, it spreads and invades different healthy tissues [5]. If a patient experiences cancer for a second time after receiving treatment for the first time, the cancer is called recurrent [6]. When the cancerous cells are not confined in a single position and expand to the surrounding regime of the body, the cancer is defined as metastatic cancer [7]. Invasive ductal carcinoma and invasive lobular cancer are commonly observed [8].

In contrast, inflammatory breast cancer, tubular breast cancer [9], colloid breast cancer [10], and metaplastic breast cancer are less common among patients. The size and location of the cancerous cells are the basic parameters for recognizing the stage of breast cancer [11–14]. Stage zero means abnormal cells are observed in a specific tissue regime. Stage I means the tumor size is below 2 cm [12]. Stage IIA means that the tumor is living in the lymph nodes in the armpit with a length of two centimeters or smaller [13]. Stage IIB means the tumor still lives inside the armpit lymph nodes with a size of 2 to 5 cm [13]. Stage IIIA means the cancer can be larger than 5 cm, but this tumor can be found in the armpit lymph nodes. Stage IIIB means the tumor is any size, and this cancer has already spread to the chest wall or breast skin [12–14]. Stage IIIC means a cancer of any size has already spread to the breastbone or around the collarbone [13]. Stage IV means the cancer has spread to distant organs [12–14].

Global cancer statistics 2020 reported that female breast cancer cases (i.e., 2.6 million) can be observed more frequently than lung cancer cases (i.e., 2.2 million) [15]. In the United States, one woman among eight women has experienced breast cancer, which has been considered the most common cancer for women worldwide [16–18]. A regular X-ray mammogram is required for 65 percent of women after age 40. Up to forty percent of breast cancer cases are not found by mammogram but rather by someone themselves, and over twelve thousand women in the United States before the age of forty are diagnosed

with breast cancer each year. Men also experience breast cancer, and they do not undergo routine mammograms. So, all of this means that understanding the signs and symptoms of breast cancer and knowing what to look and feel for are critical in finding breast cancer early before it progresses [18]. Several characteristics and symptoms need to be considered for breast self-examination (BSE). It is wise to check these signs and symptoms by lying down or standing in front of a mirror with arms above the head [17]. A doctor should be contacted if anyone notices any of these signs or other changes unrelated to menstruation, pregnancy, hormones, or other medical reasons. A thickened breast area can be a sign of breast cancer [19]. One sign is a sunken or inverted nipple. For some, inverted nipples are a typical shape of the breast, but for those new, the tumor can pull the nipple inward, flatten it, or cause it to sink [20]. If there is a dimple or ridge dimple in your skin that is not caused by tight clothing, it can be due to the tumor pulling the skin inward. This will create a divot or a dimple-like shape, and if it does not go away or if it worsens, be sure to reach out to your doctor. Another sign is a change in the breast shape or breast size [21]. It is common for women to have different sizes or shapes of breasts from the left compared to the right, and breastfeeding and menstruation can cause further fluctuations, but if your size or shape change does not connect to these issues, then you want to follow up with your doctor. In relation to nipple changes or scabs, if you see a scab-like area that is white or red over the nipple area that is not related to anything like breastfeeding, be sure to reach out to your doctor [22]. A hard lump or bump is also a common sign of breast cancer [23]. Lumps in the breasts are very common and may be a sign of natural breast tissue; they may also be something like a cyst or a benign lump, but this is why it is important to perform a monthly breast self-exam because you will get to know your breasts and know what lumps you have and what lumps may be new or growing. If there is a hard lump that is not moving or growing, a further checkup is required. A red or hot swollen breast is a common sign of infection such as mastitis from breastfeeding or from other skin changes such as eczema [24]. Further assessments are required if there is a color change due to the abnormal flow of lymphatic fluid [25]. A growing or enlarged vein on the breast can appear in response to circulatory issues, breastfeeding, or weight changes [16]. However, if a vein enlarges with other signs, such as redness or swelling, it may be a sign of breast cancer. In relation to sores on your skin, breast cancer can cause tissue changes from the inside to outside and can cause sores or wounds on the breast. You may also feel a lump with the sore if you have no known reason for a sore. This could be a sign of breast cancer, also known as Port Orange, when the skin of your breast appears dimpled, like how the peel of an orange looks; this can be a sign of breast cancer [26]. It is caused by breast swelling which makes the hair follicles appear dimpled on the skin. In relation to unexpected nipple discharge, nipple discharge is commonly related to developing breasts, infection, cysts, pregnancy, and breastfeeding, but if any of these common issues are not the cause, it can be a sign of breast cancer [16]. Itching of the breast can be a typical sign of skin changes, dryness, or even lotions or body washes that cause irritation, but if it does not go away with product changes or skin care, it may be a sign of breast cancer [27]. So, those are the main symptoms of breast cancer. Cancerous cells can develop at any age, irrespective of gender.

Screening techniques for breast cancer can help patients to detect cancer at the initial stage [28–30]. Screening can be performed in two ways: BSE and clinical breast examination (CBE) in which medical professionals examine the chest physically. In CBE, conventional breast screening methods such as mammography, ultrasound, positron emission tomography (PET), and computed tomography (CT) are employed. Mammography has a high sensitivity of 97% [31], a specificity of 64.5% [31], a positive predictive value of 89% [31], a negative predictive value of 90.9% [31], and diagnostic accuracy of 89.3% [31], but this technique has several limitations: ionizing radiation is used [32], the density of the breast affects the sensitivity and specificity [32], limited dynamic range, low-contrast and grainy image due to the sensitivity drops with tissue [31], etc. Ultrasound is not suitable for bony structure images even though this method has a sensitivity of 80% [33] and a specificity of 88.4% [33]. Ultrasound provides a low-resolution image, and an experienced operator

is required during the examination [32]. People also consider PET for screening of breast tumors. The sensitivity of PET is 68% for a tumor less than 2 cm [34] and 92% for a tumor of 2 cm to 5 cm [35], and the overall accuracy for detecting in situ carcinomas is low (sensitivity: 2–25%) [34]. PET has some disadvantages; viz., this method is an expensive method, uses ionized radiation [32], has limited resolution and slow imaging time [32], etc. CT scanning is an effective method for the early diagnosis of breast cancer due to its sensitivity (84.21%), specificity (99.3%), and accuracy (98.68%), but still, this method has limitations, i.e., successive CT scanning of radiosensitive organs like the breast can cause long-term effects [36–39]. When experts can observe initial symptoms associated with breast cancer in patients, they suggest applying an X-ray to the breast to collect an X-ray image for further observation and determination of breast cancer [30]. Sometimes, fluid-filled lymph is cancerous and is considered less dangerous than solid lymph. If there is a signature of fluid-filled or solid lymph, the breast is considered for an ultrasound checkup for further confirmation of breast cancer fill [33]. On the other hand, some specific cells from a particular regime associated with breast cancer can be removed by breast biopsy [36–39]. In that case, breast tissue is monitored under a microscope. Further investigation is performed to observe different growth factors associated with breast cancer, viz., estrogen receptor (ER) [40–42], progesterone receptor (PR) [41], human epidermal growth factor receptor (HER) 2 (HER2) [43], triple-negative breast cancer (TNBC) [44], etc. The test results associated with a biopsy can help doctors design a patient's treatment. However, all the methods mentioned above present several difficulties in generalized applications. This review article will focus on magnetic resonance imaging, microwave sensors [45], and different biosensors [46–48] in terms of mechanism, sensitivity, limit of detection, linear range, complexity, and availability for detecting early-stage breast cancer.

The rest of the paper is organized as follows: Section 2 presents different biomarkers associated with breast cancer. Section 3 highlights nuclear magnetic resonance imaging. Section 4 discusses microwave sensors for breast cancer. Section 5 explains biosensors for breast cancer. Section 6 represents challenges for existing techniques and prospects. Section 7 provides a brief conclusion.

## 2. Breast Cancer Biomarkers

Biomarkers are biological markers [49–51]. They are signals from the body that help doctors understand the impact of cancer cells on other cells. The production of biomarkers can be due to cancer itself or other cells in the body corresponding to the response of cancer cells. The information gained from biomarker testing allows doctors to create a personalized treatment plan based on the characteristics of a person's cancer [49–52]. Doctors investigate different biomarkers to evaluate cancer. Biomarkers may confirm the presence of cancer, diagnose cancer, identify the chance of a condition becoming more serious, be prognostic, or predict how a person might respond to a drug or treatment [49–51]. Biomarkers can help doctors to develop a personalized treatment plan. Biomarker testing allows the healthcare team to learn about a person's cancer. This can also be called molecular profiling, genomic testing, testing for mutations, etc. Doctors utilize biomarker test results to determine the traits of a cancer. Some biomarkers measure complex activity that occurs in the body related to the body's genes, proteins, and hormones [49–51]. Biomarkers can also measure genetic material from cancer cells in the bloodstream. Tests for these biomarkers are more specialized, and different cancer types are detected by other biomarkers [49–51]. For example, biomarkers for colorectal cancer are different from biomarkers for lung cancer. The most used breast tumor markers are ERs [52], PRs [52], HER2, TNBC [44], etc.

### 2.1. Estrogen Receptor

Estradiol circulates throughout the body in vascular circulation and freely diffuses through the cell's plasma membrane. Estrogen interacts with the ER in the plasma membrane, cytoplasm, and nucleus. Once estrogen binds to the receptor, it interacts with the immense heat shock protein HSP90 [53]. HSP90 dissociates, allowing the ER to undergo

a conformational change and dimerization, activating specific transcriptional sites on the receptor. Both activating function (AF) 1 and AF2 are activated in the ER. The dimerization ER then translocates into the nucleus and binds to the estrogen response element or ERE of DNA [53]. Co-activators and other transcription factors are recruited to the site, leading to this transcription of several estrogen-sensitive growth-promotion-associated genes via RNA polymerase. This causes further cancer cell division. Therapies that inhibit estrogen-estrogen receptor interactions can inhibit tumor growth [53].

### 2.2. Progesterone Receptor

PR is considered a hormone receptor and is governed by the ER. The interaction of PR and chromatin changes the binding position of chromatin and ER [54]. The expression of PR is dependent on estrogen. PR has been considered a prognostic biomarker associated with hormone-positive breast cancer. High expression of PR is related to tumors with a better baseline prognosis.

### 2.3. Human Epidermal Growth Factor Receptor 2

HER2 is a protein found on the surface of cells, regulating cell growth by transmitting signals from outside the cell to the nucleus inside the cell. Overexpression or an excess of HER2 receptors on a breast tumor cell or multiple copies of the HER2 gene in the cell's nucleus could lead to a particularly aggressive type of breast cancer [55]. This cancer became known as HER2-positive breast cancer. HER2 can impact a network of messages or signals inside the cell that helps tell the cell to grow, divide, or stay alive. This network is called the HER pathway [55]. Preclinical studies have shown that the HER pathway is overly active in HER-positive cancer. Too many signals running through the HER pathway can lead to uncontrolled cell growth and potentially contribute to cancer development. Receptors are a type of protein that exists on the surface of both standard and cancerous cells and are responsible for breaking signals from outside the compartment into the cell. Each receptor must dimerize with another HER receptor to send signals along the pathway [55]. Receptor pairing is thought to be critical in tumor growth and survival. However, any HER2 receptor can potentially contribute to human development and survival. The HER2 receptor may play the most significant role in cancer. HER2-containing receptor pairs lead to stronger signals through the HER pathway compared to receptor pairs that do not contain HER2. For example, the pairing of HER2 and HER3 is thought to trigger the most robust signaling of all receptor combinations [55]. HER2-positive breast cancer is often considered aggressive because it grows and spreads quickly and has a poor prognosis. HER2 overexpression affects about 25 percent of people with breast cancer.

### 2.4. Triple-Negative Breast Cancer (TNBC)

The lack of ERs, PRs, and HER2 expression is known as TNBC. TNBC is also associated with mutations in the BRCA1 gene, which is a gene linked to hereditary breast cancer. Many cells signaling pathways are dysregulated in TNBCs, such as the canonical WNT/beta-catenin signaling pathway, which controls many cellular processes like multiplication, movement, and specialization. When interrupted, this process can lead to changes in how cells replicate, leading to tumor formation. In TNBCs, the WNT/beta-catenin pathway is over-activated. When WNT binds to frizzled receptors, another protein called AXIN becomes activated too. AXIN usually is part of the destruction complex, a collection of various proteins and kinases. When this pathway is inactive, the destruction complex prevents beta-catenin from accumulating in the cell by binding to it. However, without AXIN, the destruction complex cannot bind to beta-catenin. This allows beta-catenin to collect and eventually move into the nucleus, activating other co-activator molecules. These co-activators are responsible for triggering a group of oncogenes, genes involved in tumor formation. The proteins these oncogenes produce give rise to the proliferative, invasive, and metastatic abilities characteristic of TNBC cells. Like many diseases, the treatment of TNBC is tailored to the individual diagnosed. Standard treatment methods

include surgery, chemotherapy, radiation, and immunotherapy, depending on the stage of cancer development.

There are some nucleic acids such as circulating RNA (circRNA) [56], microRNA, long noncoding RNAs (lncRNAs), circulating tumor DNA (ctDNA), circulating cell-free DNA (ccfDNA), BRAC1, and BRAC2 which have been considered as cancer biomarkers [32]. Some proteins are also considered biomarkers, viz., CD24, CD44, MUC1, etc. Sometimes cell segregation can be established inside the tumor, and several cells can detach from the tumor. These detached cells can enter the blood system and are called circulation tumor cells (CTs) [52]. The level of CTs can be considered to determine the stage of metastatic cancer, and associated individual treatment can be provided to the patient. Some exosomes are regarded as a signature of cancer. However, ER, PR, HER2, and TNBC are commonly considered biomarkers for diagnosis and treatment plans.

### 3. Nuclear Magnetic Resonance Imaging for Breast Cancer Detection

Nuclear magnetic resonance imaging (NMRI) is one of the most modern approaches for observing the details of any human organ [57–59]. A breast NMRI is an advanced imaging examination used to help the radiologist or clinician screen for breast cancer in certain high-risk populations. It is also commonly used for further evaluation of a known or suspected breast cancer. NMRI is no different than the equipment frequently used to evaluate the brain, the spine, or the joints. The only difference is in the way it is performed. For breast NMRI, the patient lies face down on the NMRI table onto a padded breast coil. Following the intravenous injection of a liquid called gadolinium or contrast, a series of highly specialized images are then obtained. The appeal of breast NMRI is that it is the most sensitive examination. That means its ability to find small early cancers is highly efficient in women with a strong family history or other genetic risk factors. We can use breast NMRI to screen for cancer before any symptoms appear. MRI can also be used prior to treatment following a diagnosis of breast cancer to map out the area of disease more accurately for the surgeons. NMRI has shown a high sensitivity of up to 93% and a high specificity of up to 97.2% [60–62].

### 4. Microwave Sensors for Breast Cancer Detection

Several research groups worldwide have been studying the use of microwaves for medical applications, especially for breast cancer screening [63–69]. Microwaves are high-frequency, non-ionizing electromagnetic radiation used in daily devices like mobile phones [68,70]. Low-power microwave technology is currently easily accessible and low-cost and poses no health risk. Complex permittivity is one of the material properties that dictates response to electromagnetic waves [71–73]. It measures how molecules dynamically deform when subjected to alternating applied electromagnetic fields. Different tissues, forming the breast's internal structures, present complex permittivity [74].

If the contrast between the permittivity of various materials is high enough, this variation in permittivity causes the incident electromagnetic wave to reflect at the boundary. The dielectric constant of a material describes the dielectric characteristics of the material. The value of the dielectric constant associated with a specific material can represent the ability to absorb energy from an electric field. The smaller the dielectric constant value, the less energy is absorbed from an electric field.

Radar-based microwave imaging (RMI) systems provide the fastest imaging inversion times. To reconstruct the image, several antennas are distributed around the breast phantom (please see the left panel of Figure 1). A low-power microwave signal is transmitted from each antenna, and the system registers and processes all the picked-up reflections [68]. The next step is to acquire the electromagnetic signals reflected from the breast for all combinations of antennas [70]. The process is repeated for different antenna heights. The medical doctor can view the reconstructed image after the signal acquisition (see the right panel of Figure 1). The algorithm computes the image for the whole volume, which can then be sliced into cuts for precise examination. Several authors considered microwave

imaging for the detection of breast cancer. In [64], the authors used nine Vivaldi antennas around the breast phantom to observe the tumor. In that case, one antenna was considered an electromagnetic wave transmitter, the remaining antennas were deemed scattered electromagnetic wave receivers, and the same process was considered for other antennas. The Microwave Radar-based Imaging Toolbox (MERIT) has been used to reconstruct an image from backscattered signals [75]. The results from this simulation suggested that this antenna array can detect a spherical tumor with a 5 mm diameter.

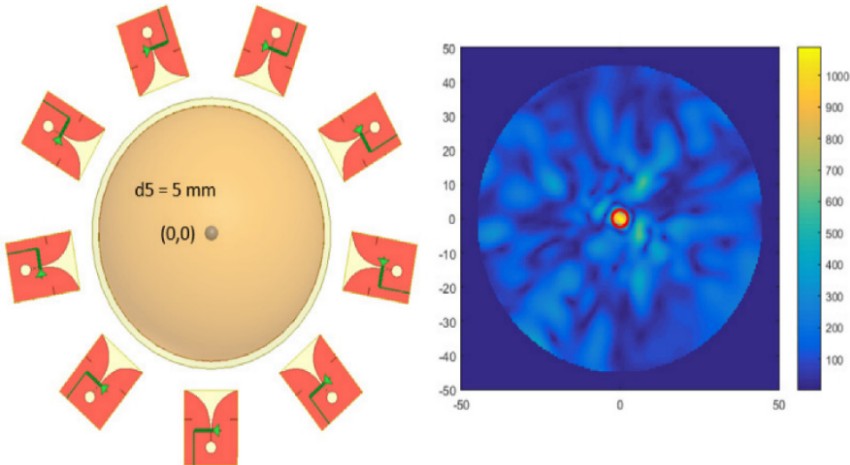

**Figure 1.** Distribution of antennas around the breast phantom (**left** panel) and detection of the tumor by considering backscattered signals (**right** panel) [64].

The breast phantom is subjected to external electromagnetic radiation, and then some portion of the energy associated with electromagnetic radiation is absorbed by the phantom. The amount of energy absorbed by the phantom is known as the specific absorption rate (SAR). This energy is responsible for raising the temperature of different phantom parts according to their dielectric properties [67]. Usually, the effective treatment area in which the temperature rises 7 to 10 degrees is considered tumor necrosis. In [67], the authors applied the microwave imaging method by considering a different number of tapered slot antennas around the breast phantom and observed that (a) the SAR at the center of the breast phantom increased with the number of antennas around the breast phantom and (b) the temperature of the tumor's regime increased by 8 to 10 degrees after considering 12 antennas. Researchers incorporated artificial intelligence for signal processing to identify patterns to classify tissues and tumors [76–79]. So, this RMI is substantially suitable for detecting breast cancer at the initial stage. Table 1 represents a comparison of different antenna elements and the detected tumor radius.

**Table 1.** Comparison of different antenna elements for detecting a breast tumor.

| Reference | Antenna Type | Number of Antennas around the Breast Phantom | Detected Tumor Radius |
|---|---|---|---|
| [64] | Vivaldi | 9 | 2.5 mm |
| [80] | Quasi Log Periodic | 16 | 5 mm |
| [81] | Monopole Microstrip | 1 | 3 mm |
| [82] | Fractal | 24 | 2 mm |
| [83] | Microstrip | 1 | 10 mm |

However, there are some complexities in generalized applications of RMI. First, the experimental breast phantom is static, while the real breast is dynamic. Second, designing an artificial breast phantom like the natural breast is challenging. There is some discrepancy in the dielectric properties of each layer of the artificial phantom compared to the real breast. Third, the time for the computation of relevant data is significantly longer than

that of other processes, and a powerful computer is required to complete the simulation. Recently, different magnetic nanoparticles have been considered along with the microwave imaging process to determine the status of a breast tumor.

Microwave-induced thermoacoustic imaging (MITI) is a modern imaging method in which electromagnetic microwave radiation is directly applied to the biological tissue to change the temperature of a specific portion of the tissue and for the generation of thermal waves by expanding and contracting a particular medium of tissue [84–86]. Finally, the specific tissue medium's expansion and contraction produces acoustic sound waves [87–89]. Several ultrasound transducers have been used to detect these acoustic sound waves and utilize them to construct an image of a specific tissue part. This method, which is a non-ionizing and non-conducting method, is simple, safe, and suitable for application in medical science, viz., early detection and diagnosis of different cancers, brain imaging [86], fetus imaging during pregnancy, and cardiovascular imaging. In MITI, microwave pulses are applied to a particular part of the tissue, and the light-matter interaction produces different physical phenomena, viz., scattering, absorption, and reflection of the light [87]. But in this method, the absorption part is considered to observe the physical properties of the tissue [87–89]. So, when the microwave pulses are applied to the tissue, a local absorption of light is established, and the temperature of the associated regime increases, ultimately generating a sound wave [87]. Several ultrasonic sensors are used to detect the acoustic sound waves. This sound wave has different physical properties, viz., amplitude, frequency, and phase. The dielectric properties of the tissue with tumors and other abnormalities lead to a rigorous change in the amplitude, frequency, and phase of the sound wave [87]. So, the resulting images can provide detailed information about the tissue. This method has a unique high resolution, is easy to apply for differentiation of normal and abnormal tissue and is very useful for the early detection of different cancers [87–89].

In [84], the authors numerically studied the change in the physical properties of both tumors and other tissues in a breast phantom. Their simulation considered an operating frequency of 2.45 GHz and an isotropic electric field produced by changing the geometrical orientation of the waveguide and tank. For their simulation and associated results, they applied different conditions to run the simulation, such as changing the tumor's size, shape, and even orientation in the breast phantom and the microwave energy (ME) pulse width and power supply. The outcomes from their simulation are as follows: (a) the breast tumor experienced more ME absorption than another part of the phantom due to the higher dielectric properties of the tumor; (b) in comparison with breast fatty tissue, the tumor can absorb more ME and can increase in temperature, even showing large pressure gradient, which leads to generating an acoustic wave, which can also be observed corresponding to the tumor and associated regimes; (c) the location of the tumor is also associated with the possibility of detecting the tumor, and the observation of tumor location in transitional tissue is more complex than the observation of tumor location in fatty tissue.

In [85], the authors considered the MITI system for detecting breast cancer in a three-dimensional framework. In this study, authors applied ME to heat the breast phantom and simulated it with different conditions, viz., tumor size, shape, location, orientation, excitation power level, and power width. The authors found the following: (a) the ME absorption by the normal and abnormal tissues is not at the same level; (b) the condensed tumor tissues absorb more electromagnetic energy than the normal breast tissue; (c) the temperature of the tumor increases more in comparison to the other breast tissues; (d) the expansion rate and the pressure variation of the tumor are higher than those of the other tissues of the breast phantom; (e) the size of the tumor is associated with the amplitude of the acoustic waves, and a bigger tumor size produces a larger amplitude of the acoustic wave; (f) tumor shape can change the temperature of the tumor; (g) greater irradiation power can increase the temperature of both the tumor and other tissue; (h) the pulse width can increase the temperature as well as the magnitude of the acoustic wave; (i) finally, the simulation can detect a tumor with a radius 0.25 cm. The MITI technique is a powerful technique for the early detection of breast tumors.

## 5. Biosensors for Breast Cancer Detection

### 5.1. Surface Plasmon Resonance Imaging

The surface plasmon resonance (SPR) method has been used to study molecular interactions in real time [90–93]. The direction of light propagation is rigorously dependent on the medium, especially on the density of the medium in which light is propagated and the associated incident angle. When light propagates to the interface of two media, the light can undergo total internal reflection due to the specific incident angle and the density of both media [93]. For a plain polarized light undergoing total internal reflection, if a thin piece of metal is placed on the top, then some refracted light is absorbed. Due to this absorption, a dark zone appears in the refracted light. This absorption of light is carried out by the electrons present in the metal. Because of the absorption of energy, the electrons start oscillating. It is not a single electron, but a group of electrons collectively absorbing the energy of light [93]. These electrons undergoing oscillation because of energy absorption are called plasmons. The refractive index (RI) associated with the medium near the metal surface is responsible for changing the surface plasmon resonant angle, and a tiny change in RI can cause a significant change in the SPR angle [93]. SPR is widely used to study the interaction between two molecules, for example, the interaction between a ligand and its receptor. The interaction between a ligand and its receptor can be easily detected just by measuring the change in SPR angle. Biomolecules on a surface interaction between DNA and RNA can also be detected [93]. Figure 2 represents the experimental setup of SPR.

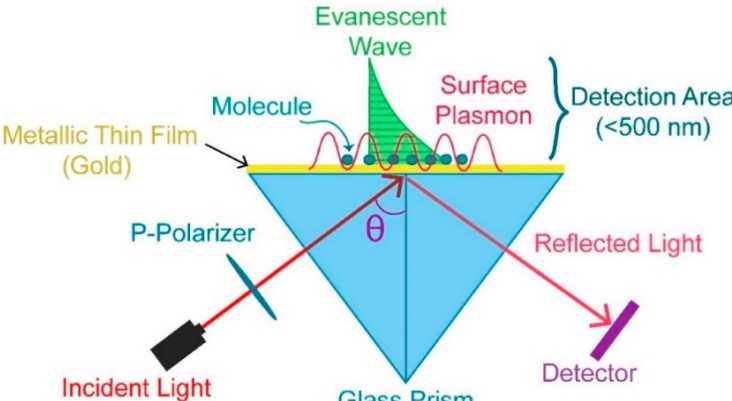

**Figure 2.** SPR experimental setup [94].

In [90], the authors developed an SPR immunosensor, in which gold nanoparticles were used to amplify the signal, to detect the ErbB2 biomarker associated with breast cancer, and they observed the following: (a) a limit of detection (LOD) of 180 pg mL$^{-1}$; (b) from breast cancer cell line graphs, it was shown that for different levels of ErbB2 (MDA-MB-436, SK-BR-3, MCF-7, etc.), the SK-BR3 cell line contains approximately 10 times more copies of the ErbB2 gene than both MDA-MB-436 and MCF-7 cell lines. In [91], the concentration of a carcinoembryonic antigen was determined by employing an SPR immunosensor, and the findings were as follows: (a) 3 μL of serum or plasma sample is required; (b) the LOD is 0.12 ng mL$^{-1}$, and the quantification limit is 0.40 ng mL$^{-1}$; (c) a linear response was observed from 0.40 ng mL$^{-1}$ to 20 ng mL$^{-1}$. This method is substantially suitable for the detection of cancer biomarkers. In [95], a photonic crystal fiber (PCF) biosensor based on SPR was developed, and the authors observed a high optical sensitivity of 6214.28 nm/RIU for the quasi-transverse magnetic mode and an optical sensitivity of 6000 nm/RIU for the quasi-transverse magnetic mode for breast cancer cells. In [96], the authors designed an SPR-based PCF biosensor to detect a cancer biomarker and found that (a) the refractive index (RI) varied from 1.392 to 1.401 for six different types of cancerous cells and (b) the maximum sensitivity was 7142.86 nm/RIU for MCF-7.

### 5.2. Colorimetric Biosensor

A colorimetric biosensor can detect the existence of different biomolecules or chemical compounds by using the concept of color change in a solution [97–99]. In a real experiment, a biosensor contains several sensing elements that can interact simultaneously with the specific analyte, viz., by breaking chemical bonds, generating new bonds, or any chemical reaction that ultimately leads to a color change in the solution or material, and these results are converted by a signal transducer into measurable data.

In [97], the authors consider HERCEPTIN as an antibody for detecting the HER2/neu receptor, one of the most remarkable biomarkers for breast cancer. For this purpose, the authors constructed a micro-calorimeter by considering micro-electromechanical technology and observed that (a) there was a peak output voltage due to the binding reaction of HERCEPTIN with HER2/neu receptor as well as the temperature difference; (b) the reaction was developed only in hot junctions; (c) the output voltage linearly (nearly) increased with HERCEPTIN quantities; (d) at the edge of the cell membrane, the intensity was dramatically increased, which means the antibody HERCEPTIN developed bonding near the cell membrane of the biomarker. In [98], the authors constructed a sandwich-type Bi2Se3-AuNPs calorimetric biosensor which has a high LOD of $10^{-18}$ M and a dynamic range of $10^{-18}$ M–$10^{-12}$ M. This calorimetric biosensor was developed under consideration of a three-step multiple signal amplification strategy for early detection of the BRCA1 mutation.

### 5.3. Surface-Enhanced Raman Spectroscopy

The Raman spectroscopy (RS) [99] technique has been employed to study the intrinsic properties of molecules and molecular mixtures in a liquid, slurry, paste, or solid phase. The vibrational transition of molecules has been considered to examine the structural properties of molecules by considering Raman spectroscopy, and a similar method has also been considered for infrared spectroscopy (IR) [100]. The basic difference between RS and IR is that RS is related to scattering while IR is related to absorption [101]. There are two types of scattering, Rayleigh, and Raman [101]. Rayleigh is elastic scattered energy, meaning the frequency of the scattered photon is the same as the excitation photon frequency. Rayleigh scattering is inadequate for finding detailed information about the chemical composition of different molecules. Ramen scattering, however, is inelastic photon scattering and is adequate for finding detailed information about the chemical composition of other molecules. RS requires two steps to occur for a molecule to Raman scatter. In the first step, the excitation-source photons excite the molecule into a virtual energy state. In the second step, the molecule relaxes by releasing scattered photons to a ground state. RS is rare, with only 1 in 10 million scattered photons being Raman scattered. Current technological advancements associated with solid-state lasers, gratings, and detectors make RS valuable for identifying and monitoring compounds in a reaction or crystallization. RS can be categorized in two ways. The first is Stokes RS, where scattered photons have higher energy than the incident exciting photons. The second is anti-Stokes RS, where scattered photons have lower energy than the incident exciting photons. Stokes RS generally has been considered for designing current Raman instruments. The interaction between the laser and the structure of the different chemical bonds among other atoms is responsible for changing the frequency of the incident photon. RS can address the specific frequency changes after the interaction of the incident laser with the chemical bond of the molecules and has been applied to recognize the bond structure of different molecules. The graph of Raman shift versus intensity associated with different chemical bonds among the molecules can express the deviation of the frequency of the scattered photon from the incident photon. The intensity associated with the Raman signal increases with the increase in the value of a chemical bond number. So, RS is a quantitative treatment for the chemical bond number. Additionally, when coupling a probe to a Raman system, measuring these bonds and their changes in real time is possible. This is valuable for understanding reaction characteristics such as kinetics initiation, endpoint, intermediates, crystal form, molecular backbone, and

mechanical information. The Raman scattering mechanism and the outcomes of experiment can be seen in Figure 3.

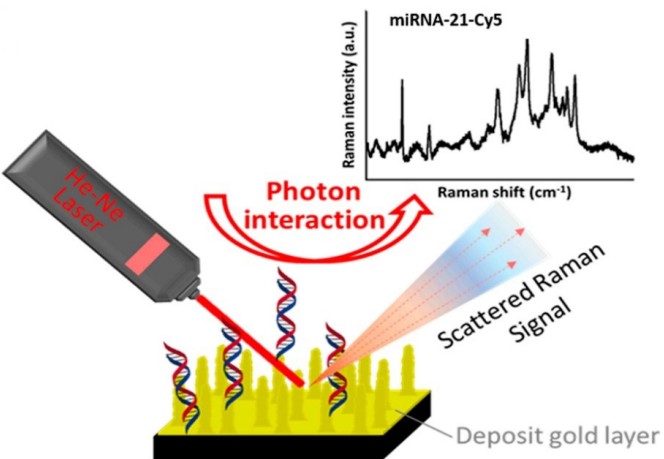

**Figure 3.** Detection of target microRNA-21 using SERS [102].

In [100], the surface-enhanced Raman scattering (SERS) technique was utilized to determine the existence of breast cancer cells/stem cells. For their experiment, the authors considered two material types, namely carbon and gold nanoparticles. The concentration of 100 µg/$1 \times 10^4$ exhibited a 5.4 increment for breast cancer cells and a 4.8-fold increment for stem cells associated with breast cancer. In [103], the authors studied the heterogeneity of HER2 associated with breast cancer. In that case, they observed four different cell lines, viz., SKBR3, MCF-7, T47D, and MDA-MB-231, from high to low HER2 expression levels. In [103], the authors suggested a functionalized silver nanostructured plate with the SERS technique, and this platform could be used for cells with different HER2 expression levels and is mostly based on the identification of the most prominent Raman peak at 1002/cm.

*5.4. Electrochemiluminescence Biosensor*

Electrochemiluminescence (ECL) is a technique in which a specific chemical reaction is triggered by applying an electrical current resulting in light emission [104–106]. This process has been used in various fields, including analytical chemistry, clinical diagnostics, and biosensors. Biosensors are analytical devices that utilize biological molecules or living organisms to detect and measure different analytes [104–106]. These devices have numerous applications, including medical diagnosis, environmental monitoring, and food safety. The use of ECL in biosensors offers several advantages over other detection techniques. Firstly, ECL has high sensitivity (i.e., 0.1 fM to 5 Pm [107]) and can detect low concentrations of analytes with high accuracy. Secondly, ECL is a fast and reliable method providing real-time results with minimal sample preparation. Lastly, ECL has a wide dynamic range and can detect a broad range of analytes, making it an ideal detection method for many biosensor applications. One of the most common biosensor applications of ECL is in the detection of biomolecules such as proteins, DNA, and antibodies. In this method, the biosensor is designed to capture the biomolecule of interest using a specific recognition element such as an antibody [104].

The captured biomolecule is labeled with an ECL probe, emitting light when an electrical current is applied. The amount of light emitted is proportional to the concentration of the biomolecule, allowing for quantitative analysis. ECL biosensors have numerous applications in clinical diagnostics, including detecting infectious diseases, cancer biomarkers, and genetic disorders. These biosensors offer a compassionate and accurate detection method that can be performed rapidly with minimal sample preparation, making them ideal for point-of-care testing. In addition to clinical diagnostics, ECL biosensors are also used in environmental monitoring to detect contaminants such as heavy metals, pesticides, and toxic chemicals. These biosensors offer a reliable and cost-effective method for monitoring

ecological pollutants, allowing for early detection and mitigation of potential hazards. In conclusion, ECL is a powerful technique that has revolutionized biosensor applications. ECL biosensors offer a sensitive and accurate detection method that can be performed rapidly with minimal sample preparation.

In [108], the authors designed an ECL biosensor by considering a silica-based mesoporous material for detecting SKBR-3 cells. The authors observed from the experimental setup that (a) the lower limit of quantitation was 20 cells/mL, (b) the linear dynamic range was 20 to 2000 cells/mL, (c) the experimental setup was able to determine the concentration of SKBR3 cells, and (d) MCF-7 and MDA-MB-231 cancer cell lines were determined. In [109], a sensing platform was designed by considering gold-nanoparticle-based carbon nitride, and this arrangement of a potential from $-1.2$ V to 0 V is associated with ECL in the presence of peroxydisulfate. The outcomes from this experimental setup are as follows: (a) the linear range was found to be from 10 fg mL$^{-1}$ to 100 ng mL$^{-1}$; (b) the LOD was 0.2 fg mL$^{-1}$; (c) the ECL emission experienced a peak at 550 nm; (d) a stable and amplified output response was observed. This ECL method is suitable for specifying cancer biomarkers.

### 5.5. Quartz Crystal Microbalance Biosensor

A crystal oscillator works on the principle of the inverse piezoelectric effect and is made up of piezoelectric material [110]. When an external voltage is applied with a specific frequency to a particular material, it produces mechanical deformation, and this material starts vibrating at the same applied frequency, which is defined as the inverse piezoelectric effect. Conversely, if we apply an external force to this piezoelectric material, it generates a voltage across the two terminals [110]. So, if we mechanically force them to vibrate at a specific frequency, they can develop an alternating current (AC) signal with the same frequency. This effect is known as the piezoelectric effect, and a material that shows this effect is known as a piezoelectric material. Different naturally occurring crystals like Rochelle salt, quartz, and tourmaline have these piezoelectric properties. Among these materials, Rochelle salt has the maximum piezoelectric activity, which means that for a given applied voltage, it generates the maximum vibration. But mechanically, this material is weak and can break very easily. Tourmaline has the most minor piezoelectric activity but is strong enough to resist breaking [110].

In [111], a CD44-biosensor was designed by following the quartz crystal microbalance (QCM) approach for breast cancer detection under consideration of different films. The experimental results are as follows: (a) the LOD for MDA-MB-231 (M231) cells was 300 mL$^{-1}$ while the LOD for MCF-7 cells was 1000 mL$^{-1}$; (b) the expression level of CD44 on M231 cells was double that on MCF-7 cells; (c) the M231 cells were less stiff than the MCF-7 cells; (d) the metastatic potential of cancer cells was rapidly detected using this method. In [112], authors applied QCM sensors to detect the biomarker CA15-3 associated with breast cancer and observed the following: (a) the one-dimensional ZnO nanostructure was appropriate for biosensor application; (b) the QCM sensor showed good stability with a response time of 10 s; (c) QCM had high sensitivity (25.34 ± 0.67 Hz/scale (1 U/mL)); (d) a linear (0.99) pattern associated with CA15.3 was observed for a range of 0.5 to 26 U/mL. In [113], the authors considered QCM for monitoring breast cancer marker CA15-3 and found that QCM sensors had high sensitivity (26. 303 ± 1.139 Hz/scale, 1 U/mL) and good linearity (0.960 ± 0.013) in the 0.5 U/mL and 100 U/mL concentration range of CA15-3.

### 5.6. Fluorescence Biosensor

If an external light is applied to fluorophores or fluorescent molecules, then the fluorophores or fluorescent molecules absorb this external light and are internally excited [114]. The excited fluorophores jump to higher unstable energy states [115]. Most of the time, the higher energy states of any system are inconsistent, and the fluorophores tend to return to the most stable lower energy states and become stable by releasing the excited energy. When the excited fluorophores tend to move the original lower stable energy states from the

higher unstable conditions, the exciting energy is emitted by a fluorophore with a different color from the original absorbing light's color. The process of absorption, excitation, jumping to higher unstable states, and becoming stable after releasing and again jumping to the lower stable energy states is known as fluorescence. In fluorescence experiments, laser light is applied to fluorophores to excite them in a correct specific range [115]. After absorbing the energy from the laser light, these fluorophores jump to the higher energy states and again to the lower stable energy states. Finally, the detector detects the corresponding emitted light from the fluorophores [115]. In biological experiments, fluorescent dyes or antibodies have been used to detect the structure and processes of proteins, tissues, organisms, etc. The existence and number of fluorescent molecules can be determined using microscopy or flow cytometry techniques [115]. The fluorescence method is regularly considered due to its ultra-high sensitivity and process producing a quantitative result. DAPI, which has been used to detect DNA, and ethidium bromide, which has been used to intercalate DNA in agarose gels, are common fluorescent molecules. Green fluorescent protein can be found in jellyfish. A biological response has been converted to a measurable signal by a biosensor by combining enzymes or antibodies with a transducer. In that case, fluorescence molecules are considered as the transducer in fluorescence biosensors to produce a measurable signal.

In [116], the authors developed a complex 3D polymeric network, called a hydrogel, by combining carbon dots for early detection of cancer cells. For this purpose, the authors fabricated a nanocomposite hydrogel and functionalized it with an ssDNA probe to detect human breast cancer MCF-7 cell lines. In that case, they considered the DNA hydrogel bioassay strategy. They observed fluorescence spectra for three nanocomposite hydrogels (viz., GA-CDs-CH, NB-CDs-CH, and NB-CDs-CH) for a different concentration of microRNA-21. For a particular condition, the decreased fluorescence intensity, and the logarithm of microRNA-21 concentration provided a linear relationship at 0.1 to 125 fM for GA-CDs-CH and NB-CDs-CH hydrogel and 0.1 to 26.3 fM for B-CDs-CH hydrogel. These nanocomposite hydrogels are applicable for multicolor imaging of MCF-7 cancer cells. In [117], the authors employed the fluorescence emission method to detect biomarkers (microRNAs, miR-21, miR-195, and miR-155). They considered label- and enzyme-free methods and fabricated a fluorescence reporter based on DNA-templated copper nanoclusters to detect biomarkers. They found a linearity range of 500 nM to 3 μM for a detection limit of 1.7pM. This fluorescent spectrum method is suitable for detecting microRNAs without any linkers and enzymes.

### 5.7. Flow Cytometry

In the flow cytometry method, the physical scattering process is considered to determine the size, shape, and properties of a single cell in a combination of different cells [118–120]. Initially, the sample is mixed with saline solution in the cytometer and allowed to pass through a narrowing channel which finally causes a single cell to live in a single droplet. When this droplet with a single cell passes through the laser pulse, the droplet/cell interacts with the laser pulse [118]. The point at which the cell interacts with the laser pulse is known as the "interrogation point". After the laser pulse interacts with the droplet/cell, it scatters in different directions. The portion of the laser that moves forward in focus after scattering is known as forward scatter (FSC) [118–120], and the portion of the laser that moves in the sideways direction after scattering is known as side scatter (SSC) [118]. Experimental setup of flow cytometry can be seen in Figure 4.

Forward scattering pulses provide information associated with the size of a cell, while sideways scattering pulses provide information associated with the shape and internal complexity of a cell [118]. The forward (sideways) scattering laser pulses are detected by detector 1 (detector 2) and converted into voltage pulses. These data are converted to a histogram plot according to the scattered light and cell number [118]. Researchers investigate both forward and sideways scattering data together and determine a cell's size, shape, and complexity [118–120].

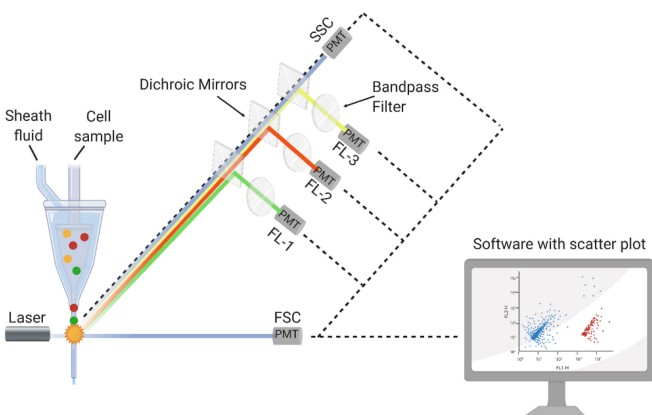

**Figure 4.** Experimental setup of flow cytometry [121].

In [122], the authors applied the flow cytometric method to determine the number of cancer cells in peripheral blood. They studied the reactivity of several reagents, viz., anti-CK20, anti-CK7, anti-pan-CK, anti-CK8, anti-CK8/CK18, and anti-CK18, and found that this sensing system could detect a single cancer cell from $10^6$ to $10^7$ nucleated blood cells. In [123], authors saw different types of breast cancer markers, viz., ER, PR, HER-2/neu, (EGFR), and E-cadherin, by applying the flow cytometry method and reported the following: (a) a rigorous positive pattern had been observed for E-cadherin with an 85–92% marker expression; (b) in comparison with E-cadherin, a weakly positive expression pattern had been demonstrated by ER, PR, and HER-2/neu; (c) there was an inverse relation between HER-2/neu expression levels with MCF7 cell lines. So, flow cytometry is a powerful tool for detecting breast cancer in the primary stage.

### 5.8. Microfluidic Chip

Fluid can flow in two different ways, viz., laminar flow and turbulent flow. In laminar flow, the streams are parallel [124]. So, there is no mixing between the streams. Then there is turbulent flow, where various inertial forces cause the fluids to move randomly. Turbulent flows tend to affect reading accuracy [124]. In the micrometer regime, the liquid flow is to be considered as laminar flow. So, a complex pattern is produced by the joining points of two fluid streams in a microchannel, and diffusion is required to explain this.

On the other hand, the scaling law is a comprehensive way of describing the variation in the physical quantity as a function of the size of the system [124]. Microfluidics is a study associated with submicron fluidic structures, and this knowledge is applied to developing lab-on-chip or microfluidic chip technology [124], which combines different operations. All kinds of operations are completed in a micron-sized chip. Microfluidic chips use semiconductor-like micro-electromechanical processing technology to build a microfluidic system on the chip [124]. Different interconnected paths and liquid phase chambers exist in the chip, and the analysis process associated with the experiment is printed over the chip [124]. Biological samples with their solutions have been loaded, and different pumps and forms of electro-osmotic flow have been used to transmit the flow of the buffer in the chip to form a micro flow path. So, the flow experiences several successive reactions over the traveling trajectory. For accurate and precise sample analysis, different detection systems, along with analysis methods, have been employed in microfluidic chips [124]. A complete microfluidic system includes many components, mainly micr-pumps, microchips, microvalves, and detection units. A micropump is the power source for microfluidic transportation [124]. Microchips can also be classified according to different dimensions and according to the material of the chip. They can be divided into silicon, glass, organic polymer materials, and paper materials [124]. Based on the types of channel arrays, they are single-channel and multi-channel chips. According to the network shape, they can be divided into linear, spiral, curved snake, polygon, folded, etc.

In microfluidic technology, a small chip can integrate several steps associated with miniaturing and automation. There is high throughput since microfluidics can be designed as multiple channels, and multiple items can be tested in parallel on the same sample as needed. The detection time is significantly shortened, the efficiency is improved, and high-throughput detection is realized. The reaction unit cavity associated with the microfluidic chip is tiny. The reagent usage is far lower than that of conventional methods, and thus, the consumption of reagents is greatly reduced. The demand for sample sizes is small. Since the task is completed on a small chip, the sample size required to be tested is very small, so microliters or even nanoscale amounts could be enough [124].

Biological analysis based on microfluidics includes microfluidic-based analysis in cell mode and microfluidic-based exosome separation analysis [125]. Clinical diagnostic research based on microfluidics includes pathogen detection and analysis based on microfluidics. Cancer detection and analysis can be based on microfluidics. Food safety analysis, including detecting pesticide residuals, pathogenic bacteria, heavy metals, and food additives, has been based on microfluidics [126]. Environmental monitoring and analysis have been based on microfluidics detection methods including electrochemical, optical, and MS methods. Creative Biolabs is a leading customer service provider in cancer next-generation sequencing with a professional platform and rich experience. Integrated microfluidic devices could differentiate breast cancer patients from healthy controls with high sensitivity (90%) and specificity (>95%), making this strategy applicable to the analysis of clinical specimens [127].

In [128], the authors considered a glass substrate and polydimethylsiloxane for constructing a microfluidic device applicable to quantifying the circulating exosomes. They collected and measured the circulating EpCAM-positive exosomes from six breast cancer patients and three healthy people and observed that the level of EpCAM-positive exosomes significantly increased in breast cancer patients compared to healthy people. Personalized breast cancer therapy is associated with molecular classification by HER2. The authors considered nineteen breast cancer patients with HER2-positive exosomes. They utilized a microfluidic device to discover the secret HER2-positive exosomes in breast cancer cells. They observed that the level of HER2-positive exosomes in SK-BR-3 CM is significantly higher than that in both MCF7 and M231 CM. In [129], the author designed a detection platform for miRNA biomarkers by considering a microfluidic chip and observed that the detection limit was 1 pM and the time was 30 min. This microfluid chip approach is suitable for detecting breast cancer biomarkers and quantifying the amount of existence.

### 5.9. Molecularly Imprinted Polymer

A molecularly imprinted polymer (MIP) is a polymer that has been designed by applying the molecular imprinting technique, and MPI can generate cavities in the matrix of the polymer to attract a particular template molecule [130]. Manufacturing a nano-MIP involves controlled polymerization around a targeted choice, immobilized on solid-face support [131]. This is then packed as a column. The nano-MIP is made by first exposing the target to monomers that find their optimal binding positions on the target through self-assembly. Different solvents and cross-linking agents have been used to complete this step. In the pre-polymerization step, the distribution of the functional monomers around the template is precisely organized by the chemical properties and shape (see Figure 5).

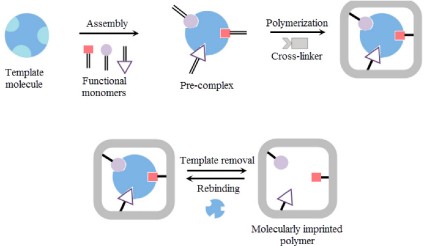

**Figure 5.** Different steps of molecularly imprinted polymer preparation [132].

The monomers are fused together through polymerization by exposing the column to UV light, activating a catalyst that generates free radicals. The free radicals collide with the carbon-to-carbon double bonds in the monomers, initiating a polymerization chain reaction. Once sufficient polymerization has occurred to form the desired level of polymer formation around an individual template molecule, we stop polymerization. Stopping polymerization is simple. The UV lamp is turned off, and the living radical becomes an inhibitor, mopping up the free radicals. Each discrete single polymer particle or nano-MIP has one binding site. We then wash away the unreacted monomers and lower-affinity polymers. By controlling the stringency of the wash process, we regulate the affinity of the nano-MIPs, which are subsequently harvested. We can further modify the surface of the nano-MIPs retained on the column by attaching other compounds of interest, such as biotin or peg. This allows us to create the best possible MIPs for application in a new world of diagnostic solutions. In [133], the authors observed that the sensitivity of the ammonia sensor developed using the MIP-doped boric acid method was 0.243 Hz/ppm, which is three times greater than that of a sensor developed without the MIP method (i.e., 0.079 Hz/ppm).

In [134], carbohydrate antigen 15-3 (CA 15-3) was detected by employing the MIP strategy, and the following observations were made: (a) the redox peak current decreased with an increase in the value of the CA 15-3 concentration; (b) LOD was observed 0.10 U mL$^{-1}$; (c) the MIP biosensor detected the CA 15-3 associated with breast cancer below the clinically recognized value. In [131], the author considered the MIP method to observe the susceptible BRCA-1 gene associated with breast cancer and reported that the suggested biosensor maintained a wide linear range from 10 fM to 100 nM and had an LOD of 2.53 fM.

### 5.10. Field-Effect Biosensor

Semiconductor-based biosensors (SBBs) can be directly applied to extract information associated with different cancers from biological experiments [135,136]. These SBBs have been designed to observe a sample's electrical property variation. The results that are found from these experiments are used to determine the status of the cancer and diagnose the condition. A field-effect transistor (FET) is a semiconductor device with three terminals: source, gate, and drain. The FET controls the conductivity between the drain and source by considering the electric field or the application of a voltage to the gate [135]. A field-effect biosensor (FEB) is a device manufactured by following the principle of the FET that can detect the existence of biological molecules, such as proteins or DNA, in a sample [137–139]. Currently, several FEBs are available, viz., ion-sensitive field-effect transistors, metal-oxide-semiconductor field-effect transistors, organic field-effect transistors, and nanowire field-effect transistors [140]. Currently, ultra-sensitive nanowires can be utilized for developing lab-on-a-chip devices. The epitaxial method has been employed by considering nanowires for designing three–five SBBs. In that case, a metal droplet is deposited over the surface of the semiconductor, and nanowires vertically grow from the metal droplets [140–142]. The nanowire behaves like a junction, and due to the ultra-sensitive properties of nanowires, it can easily capture and transmit the optical and electrical signals to the next portion of the device. Nanowires are so thin that the free charges exist at the nanowires' surface [140–142]. So, when target molecules contact the nanowires, the surface free charges from the nanowires bond to the target molecules, and a change in the nanowires' surface charge density is observed. So, the interaction between the nanowires and target molecules can change the surface charge density of the nanowires, and this change in the charge density can indicate the existence of different biomarkers [140–142]. So, the idea of FET is an ultra-sensitive approach to recognizing the existence of specific biomarkers, and a carbon nanotube (CNT) FET device with high sensitivity down to the fM range can be regenerated in an acidic environment, enabling the reusability of the sensor device [140–142].

In [135], the author considered a DNA-functionalized CNT-based FET biosensor in which gold nanoparticles are considered linkers to anchor probe molecules to detect breast cancer exosomal miRNA. Initially, exosomes are collected from the breast, and specific

miR21 is extracted from the exosomes. Hybridizing the target miRNA and immobilized DNA probe establishes a current flow. So, before and after the hybridization, the variation in the current flow can predict the existence of exosomal miRNA21. The CNT-based FET biosensor can directly convert the hybridization of biomolecules with a sensing interface into an electrical signal.

In [136], a single-nanotube FET array was considered to detect a change in the electrical signals due to the development of a bond between the receptor proteins and their corresponding antibodies. In that antibody device experiment, the authors considered two antibodies (IGF1R and Her2) at different levels. They established an interaction with intact MCF7 and BT474 collected from human breast cancer cells, and a 60% conductivity decrease was observed when BT474 or MCF7 breast cancer cells interacted with IGF1R and Her2 antibodies. The antigen-antibody interaction changed the system's free energy, increased the band gap, and decreased the system's conductivity. A single cell with a diameter 1000 times smaller than a cancer cell produced a bond with a single-nanotube FET, changing the conductivity.

*5.11. Electrochemical Impedance Spectroscopy*

Electrochemical impedance spectroscopy (EIS) applies an external sinusoidal potential/current to an electrochemical system and measures the corresponding outcome sinusoidal potential/current [143–145]. The applied potential has several features, viz., amplitude, angular frequency, and time dependency, and the corresponding output current has similar characteristics. However, the output current experiences a phase shift from the input external potential [144]. So, to reiterate, the input and output signals should have the same angular frequency, but a phase shift or a phase angle may offset the output signal. A complete EIS experiment applies a sinusoidal potential with different frequencies instead of a single frequency, and corresponding output currents are measured. So, a frequency spectrum is observed instead of a single frequency. Fourier transform has been utilized to convert all input and output signals from the time domain to the frequency domain. The experimental setup is a frequency-dependent system, so the impedance is calculated by calculating the ratio of the frequency-dependent potential over the frequency-dependent current [145]. The phase angle and impedance are a function of frequency, and the associated plot is known as a body plot. Sometimes, a Nyquist plot is used to show the real and imaginary impedance. The time scale is a critical parameter in electrochemical processes [145], and in most cases, the time scale associated with different mechanisms cannot be recognized globally but can be recognized locally. For example, double-layer charging can occur in a few microseconds while diffusion can happen in a time scale of hundreds of milliseconds [144]. These two mechanisms have two distinct time scales for completion, but in electrochemistry, these two-time scales are not distinguishable.

For example, in a DC voltammetry experiment, a step potential or a linear sweep is applied, and the corresponding measured current is the combination of different electrochemical processes occurring within a specific time. In that case, it is not easy to differentiate the current related to the double-layer charging or diffusion. So, DC voltammetry cannot provide a detailed picture of different mechanisms and their associated development time. On the other hand, EIS, in which oscillating perturbations are applied and corresponding currents are measured, can provide detailed pictures of individual internal mechanisms of the electrochemical reaction, and this EIS method is able to discern different electrochemical phenomena because they occur at different frequencies. EIS is widely used for direct monitoring of antibody-antigen connected reactions, DNA hybridization, and enzyme reactions, and the sensitivity of EIS can reach up to μg/mL to pg/mL [146,147].

In [143], the authors employed the impedance spectroscopy method and differentiated early-stage, invasive, and metastasized human breast cancer lines MCF-7, MDA-MB-231, and MDA-MB-435, respectively, from the standard breast tissue cell line MCF-10A. They measured the impedance for both normal breast cells and three other cancerous cell lines over a frequency range of 100 Hz to 3.0 MHz and observed the following: (a) the magnitude

of the impedance decreased with frequency; (b) the phase of the impedance initially decreased and further increased with frequency; (c) there was a significant difference between the standard cell line (for both magnitude and phase of the impedance) and the other three cancerous cell lines; (d) the magnitude and phase of the impedance decreased (increased) following the order MCF-10A, MCF-7, MDA-MB-231, and MDA-MB-435. In [148], authors identified the spectroscopic patterns for three different pathologic stages, viz., the early stage (MCF-7), invasive phase (MDA-MB-231), and metastasis (SK-BR-3), by employing bioimpedance spectroscopy under the consideration of magnetic nanoparticles. In the alpha dispersion bandwidth, the magnitude of electrical impedance spectra maintains a substantial difference for different cancer cell lines. So, impedance spectroscopy is suitable for identifying low concentrations of cancer cells.

HER2 is considered one of the most important biomarkers for identifying the proliferation of breast cancer. In [149], authors designed an electrochemical immunosensor for analyzing the HER2 extracellular domain (ECD) in human serum, and a screen-printed carbon electrode was considered as a transducer surface in the presence of gold nanoparticles. The linear sweep voltammetry (LSV) method was applied to analyze the enzymatic reaction and the antibody-antigen interaction. They observe the following: (a) the quantification and the patient's response to the therapy of the biomarker were determined for a wide range of concentrations; (b) the dilute sample with buffer was sufficient for a high HER2 ECD concentration; (c) the LOD was determined at 4.4 ng/mL; (d) the cut-off value was 15 ng/mL, so the LOD, as well as the diagnosis of HER2, was easily determined for the patients.

### 5.12. Light-Based Breast Cancer Techniques

Light can interact with the matter of different refractive indices, and the reflection, absorption, polarization, and scattering of the light can be monitored to explore the characteristics of the tissue, cells, or biological materials [150]. We have already discussed different optical biosensors, viz., SPR, SERS, fluorescence, and flow cytometry, in which physical properties of light have been considered to detect breast cancer. Several authors considered the frequency-domain photon migration (FDPM) [151–153] method for detecting breast cancer. FDPM performs content-based analysis by deriving scattering and absorption coefficients from amplitude and phase differences in backscattered, broadband-modulated (10 MHz to 1 GHz) near-infrared light [154]. Terahertz (THz) imaging is another platform for detecting breast cancer at early stages. A THz wave (0.1–10 THz) has high sensitivity to water molecules [155], is non-ionizing [156], produces low power levels [156], and has good penetration capabilities and resolution. In [155], the author demonstrated THz reflection imaging and distinguished 10 μm heterogeneous tissue sections from cancer and noncancer breast tissue. In [156], TPS Spectra 3000 pulsed THz imaging and spectroscopy were considered, and it was found that tumor detection is accurate to depths of over 1 mm.

### 5.13. Analytical Techniques Based on Nanomaterials for the Detection of Breast Cancer Biomarkers

There are different nanomaterials like carbon nanotubes, graphene oxide, quantum dots, multifarious composites, and noble metal nanomaterials (Au NPs, Ag NPs) which have been considered as a part of different biosensors to improve the detection performance of biosensors [157]. Nanomaterials are hundreds of times smaller than human cells. So, the size and the quantum tunneling effect of the nanomaterials significantly improve the performance of biosensors. In [158], the authors constructed a CdS/ZnO nanorod array-based photoelectrochemical sensor to detect MCF-7 and determined the concentration from 50 to $1.0 \times 10^6$ cells/mL, and the LOD was found to be 10 cells/mL. In [159], the authors detected the breast cancer biomarker microRNA-21 (miR-21) by applying an ECL technique with a gold-silver nanocluster supramolecular network (AuAg NCs) and reported that the linear range was 10 aM–10 pM, and the LOD was 4.5 aM. Nanomaterials are also considered for SPR [160,161], SERS [162], and colorimetry [163–165], etc.

## 6. Challenges for Existing Techniques and Prospects

NMRI pictures can express a minimal change inside a medium. Sometimes doctors suggest the consideration of the breast under an MRI experimental setup and observe a small change associated with breast cancer to determine cancerous cells further. NMRI can successfully select a shift inside the breast even though the mammogram cannot do that [57–59]. NMRI is suitable for early breast cancer detection, but the availability of NMRI is challenging in different parts of world. Sometimes, patients feel uncomfortable with NMRI due to its relation to nuclear technology. Most of these methods are cost-effective, time-consuming, and unavailable worldwide. Again, this method is hard to manage for early detection of breast cancer or regular checkups for cancerous cells.

Microwave imaging is considered an effective method for early breast cancer detection. However, these methods have some limitations. The dielectric properties of the breast phantom are not precisely measured. The contrast between the tumor and other tissues' dielectric properties is minimal for detecting the existence of a tumor inside the breast specifically. Antennas are the most critical components in microwave imaging systems. The mutual coupling between these antennas also increases with the increasing antenna numbers of a microwave imaging system [45]. This mutual interaction of the antenna's radiation further leads to a low-resolution tumor position and size image. On the other hand, a lot of data are associated with different functions and orientations of the antennas. So, it is essential to utilize a specific method to determine the physical properties of individual data related to simulation. So, in that case, we again face a significant challenge in encapsulating extensive data and defining a specific binary decision corresponding to this large dataset. Several researchers applied the concept of artificial intelligence or neural networks to certify individual datasets and predict the future of breast cancer [166–174]. So, artificial intelligence is one of the most promising tools for generalizing the problem for individuals' datasets and determining the corresponding possibilities.

In MITI, the scattering between the incident electromagnetic wave and living tissue reduces the intensity of the incident light, and the absorption of electromagnetic waves by the living tissue sometimes causes deformation of the cells. On the other hand, the temperature variation and the sound wave associated with electromagnetic phenomena ultimately provide an indication of a tumor in a regime instead of a specific point.

SPR has many advantages; viz., it is a level-free method, and potential leveling artifacts are not required. It is a direct method because it measures the binding of the actual analyte. This method can provide information about the evolution of the experiment in real time. However, SPR has some disadvantages. SPR is designed for mass change, decreasing with the distance from the surface. The LOD is around 200 nanometers, and this method is also dependent on the material, especially gold and silver. Highly viscous materials cannot be considered for an experiment, and the sample must be homogeneous to be used in the investigation. Sample preparation and probe attachment to the metal surface are challenging.

A colorimetric biosensor is a fast, inexpensive, and portable detection device that can be used for the early detection of cancer biomarkers. However, this process has trouble determining the individual components of a mixture. Chemometric methods have been considered for encapsulating large datasets of red, green, and blue values [175]. Moreover, challenges are also associated with the reproducibility of imaging and printing.

SERS can be applied to find the information of a submicron system. SERS can determine the bonding information associated with different phases of materials. Generally, this experiment can be performed in ambient conditions and can study the dynamic process. Transparent or semi-transparent materials can be considered for SERS treatment within the visible spectrum for detailed bonding information. There is freedom in the experimental setup to visualize a wide range of in situ capabilities, viz., temperature, environmental control, strain, chemical monitoring, etc. However, SERS is not free from the disadvantages. A laser is applied to the sample in SERS treatment, and the exciting laser selection is essential for extracting the bonding information because the laser is also sometimes responsible for

degrading the material. RS is also a low-probability light interaction. Our ability to detect light has improved, and the limited detection of most materials is around 1%.

ECL is a non-radioactive, flexible, and fast process for detection. However, the complexities of applying this method are related to maintaining a favorable temperature and stable electrodes and the cross-interaction between the co-reactant and the luminophore. A quencher can be introduced further to amplify the intensity of ECL [176]. An ideal luminophore, which has a lower energy band gap, which means little energy is required to make an electron transition from the valance band to the conduction band, can be considered for ECL to improve the performance of ECL [176]. Modern quantum materials are used to design these highly efficient luminophores.

There are several advantages to considering a QCM biosensor. A QCM biosensor has stable, high-sensitivity, and level-free detection capability [177]. So, the possible interference associated with the actual binding process can be ignored in this approach [177]. However, a QCM biosensor has some limitations. First, if there is non-uniform mass distribution on the quartz crystal, this leads to a frequency discrepancy. Second, the solution leak enhances the capacitive background current. Third, a minimum time, the upper limit on the scan rate, is required to locate the resonant frequency. Fourth, the QCM frequency is linearly dependent on the temperature [178]. The temperature variation can change the experimental setup. Fifth, there is noise due to the vibrating nature of the crystal. The electrochemical signal must be separated from the exciting signal.

In most fluorescence biosensors, the fluorescence intensity is considered for detecting impurities. The instrument's condition and environmental factors can easily perturb the outcome fluorescence profile. In that case, the self-calibration of the two fluorescent emission bands has been addressed for ratiometric sensors to minimize the external effects on the fluorescence intensity profile [179,180]. In fluorescent resonance energy transfer (FRET) based ratiometric sensors, the ratio of two fluorescent emission bands is not affected by the concentration and fluctuation of the source [181]. Conjugated polymers are sometimes considered for delocalizing electrons and light harvesting, ultimately amplifying the fluorescence intensity output profile [181].

Flow cytometry is an optical setup to observe individual cell properties to differentiate abnormal cells from normal ones. In that case, the wavelength of the incident laser light is comparable to the size of the cell. Otherwise, the light-matter interaction will not provide detailed information about a single cell. On the other hand, this method can generate colossal data by considering a single cell, and this massive information can be categorized by employing the knowledge of machine learning [166–171] to determine the binary outcomes and corresponding possibilities.

Microfluidic biosensors are more compatible than non-microfluidic biosensors and can produce both quantitative and qualitative results within 3 h. The LOD for microfluidic biosensors is $3 \times 10^3$ CFU/mL, while the LOD for non-microfluidic biosensors is $3 \times 10^4$ CFU/mL. However, microfluidic biosensors have a smaller dimension and high surface-area-to-volume ratios, which ultimately leads to a lower flow rate and a high-pressure difference over a small distance.

The Debye shielding effect, which can be observed in any FET-based biosensor, is considered to reduce the LOD. The electrostatic field of the surface charges associated with biomolecules will vanish after the Debye screening length [182]. The number of accounts near the surface of the biomolecule and the same number of opposite directions also exist around the biomolecules. These opposite charges can materialize the electric field due to surface charges. In a buffer solution, the electrostatic field exponentially decreases and becomes zero after a particular distance, and this critical distance is known as Debye screening length. In FEB, the LOD is rigorously dependent on the Debye length limitation [182]. The sensitivity of an FET-based biosensor is significantly reliant on the strength of the ions.

Biosensors face several challenges in determining the status of a biomarker in a sample. For example, a lower concentration of biomarkers is one of the most critical factors. In that

case, a lower biomarker concentration produces weak electrochemical interactions and associated weak signals, which most biosensors have difficulty detecting. In that case, several secondary antibodies or antibody-like substances are to be considered for amplifying the electrochemical signal [52]. On the other hand, the interference between different meanings generates different signals/noises. To resolve these complexities, several ideas have been considered in sensing techniques. Examples include the following: (a) quantum nanomaterials have been considered as an amplifier of the weak primary signals; (b) secondary antibodies can be introduced to amplify the raw signals; (c) secondary/indirect and larger indirect signals can be detected, and these secondary/indirect signals can be decomposed to observe the properties of primary/direct signals [52]. The sensitivity of a biosensor could be improved under the consideration of detection strategies for secondary/indirect signals. Background interference can be minimized if the sample undergoes a pretreatment. For MIP, it can be considered that biomolecules have difficulty coping with harsh environments.

Finally, different biomarkers can simultaneously exist in a single sample, but most biosensors can detect one at any time. Table 2 represents the linear range and the limit of detection of different biosensors for different biomarkers.

**Table 2.** A summary of linear range and limit of detection of different biosensors.

| Type of Biosensor | Target | Linear Range | Limit of Detection | References |
|---|---|---|---|---|
| SPR | HER2 | $10^{-12}$–$10^{-6}$ g/mL | $9.3 \times 10^{-9}$/mL | [183] |
| | MCF-7 | $10^4$–$10^6$ cells/mL | 500 cells/mL | [184] |
| Calorimetric | BRCA1 | 1 fM–100 pM | 0.34 fM | [99] |
| | BRCA1 | $10^{-12}$ M–$10^{-18}$ M | $10^{-18}$ M | [98] |
| SERS | CA153 | - | 0.0001 U/mL | [185] |
| ECL | HER2 | 0–900 pg/mL | 20.4 pg/mL | [186] |
| | MCF-7 | 10–100 cells | 10 cells | [187] |
| | CA19-9 | 0.0001–10 U/mL | 0.000046 U/mL | [188] |
| | miRNA-21 | 100 aM–100 pM | 19.05 aM | [189] |
| | CA19-9 | 0.0005–150 U/mL | 0.0002 U/mL | [190] |
| | MUC1 | 1 fg/mL–1 ng/mL | 0.62 fg/mL | [191] |
| | HER2 | 1 fM–1.0 nM | 1 fM | [192] |
| QCM | MCF-7 | $10^2$–$10^5$ cells/mL | 430 cells/mL | [193] |
| | MCF-7 | $10^2$–$10^7$ cells/mL | 32 cells/mL | [194] |
| | MDA MB 231 | - | 12 cells/mL | [195] |
| Fluorescence | CA125 | $10^{-2}$–127 U/mL | $10^{-4}$ U/mL | [196] |
| | CYP1A1 | - | 1 pg/mL | [197] |
| | miRNA-21 | 1 pM–10 nM | 55 fM | [198] |
| | miRNA-21 | - | 18.7 pM | [199] |
| Microfluidic | HER2 | 1 fM–100 nM | 1 pM | [200] |
| | CA15-3 | 10–1000 μU/mL | 6.0 μU/mL | [201] |
| | CEA | - | 0.2 ng/mL | [202] |
| | Glypican-1 | - | 10 exosome/μL | [203] |
| | PTK7 | - | 0.4 nM | [204] |
| MIP | CA 15-3 | 0.1–100 U/mL | 0.10 U/mL | [134] |
| | HER2 | 1–200 ng/mL | 0.43 ng/mL | [205] |
| | CA 15-3 | 5–50 U/mL | 1.5 U/mL | [206] |
| Field effect | miRNA-155 | 0.1 fM–10 nM | 0.03 fM | [207] |
| | HER2 | 100 pM–1.0 μM | 1.0 pM | [208] |
| | EGFR | 10 fM–10 nM | 10 fM | [209] |
| | CA125 | $10^{-9}$–1.0 U/mL | $5 \times 10^{-10}$ U/mL | [210] |
| EIS | BRCA1 | 50 fM–1.0 nM | 1.72 fM | [211] |
| | HER2 | 1 pg/mL–100 ng/mL | 172 pg/mL | [212] |
| | miRNA-155 | 10 aM–1.0 nM | 5.7 aM | [213] |
| | miRNA-21 | 10 fM–100 pM | 4.3 fM | [214] |
| | EGFR | 1 pg/mL–1 μg/mL | 0.88 pg/mL | [215] |
| | HER2 | 0–4.0 μg/L | 6.0 μg/L | [216] |

Determining different biomarkers in a sample at a specific time is to be considered a primary challenge. In that case, several biosensors can be considered for detecting different biomarkers in the same sample at a specific time. This challenge can be resolved if several techniques are considered to design a complex biosensor that has high sensitivity, specificity, and a simple operation procedure and can simultaneously detect multiple targets. This complex biosensor can be designed under two considerations: (a) the strategy, by which biomarkers are to be detected, should include improving the sensitivity and enhancing the specificity, and (b) the detection equipment should be easy to operate and able to complete multi-target detection simultaneously. For developing a complex biosensor, the mechanism of electrochemical and optical biosensors can be combined with the mechanism of microfluidic chips to achieve better performance. Most of the microfluidic-chip-based biosensors are innovative in devices, while electrochemical and optical biosensors are more innovative in terms of detection strategies and materials.

## 7. Conclusions

Breast cancer is a global problem. Detection of breast cancer at an early stage is significantly important for designing individual treatment plans. Biomarkers that are the primary signatures of breast cancer and a detailed discussion about the biomarkers associated with breast cancer have been highlighted. There are two ways, i.e., BSE and CBE, by which breast cancer screening can be performed. This review article reported various approaches for BSE as well as different techniques (viz., mammography, ultrasound, PET, CT, NMRI, RMI, MITI, SPR, colorimetric, SERS, ECL, QCM, fluorescence, flow cytometry, microfluidic chip, MIP, field-effect, EIS, light-based, and nanomaterial-based analytic methods) associated with CBS for the early detection of biomarkers. Several parameters, i.e., sensitivity, specificity, accuracy, LOD, linear range, tumor radius, and temperature variation, have been considered for comparing different techniques. This review also addressed the existing challenges for detecting biomarkers, such as sensitivity, specificity, weak signal, background interference, and detecting different biomarkers at a specific time, and provided different solutions to resolve these challenges.

**Author Contributions:** Conceptualization, L.W. and N.A.C.; formal analysis, L.W., N.A.C., L.G. and M.K.; writing—original draft preparation, N.A.C.; writing—review and editing, L.W., N.A.C., L.G. and M.K.; project administration, L.W., L.G. and M.K. All authors have read and agreed to the published version of the manuscript.

**Funding:** This research received no external funding.

**Institutional Review Board Statement:** Not applicable.

**Informed Consent Statement:** Not applicable.

**Data Availability Statement:** All data are presented inside the manuscript.

**Acknowledgments:** The authors are grateful to anonymous reviewers for their constructive suggestions which have significantly improved the quality of our manuscript.

**Conflicts of Interest:** The authors declare no conflict of interest.

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
