# Peer review of "Exploring the Potential of Sensing for Breast Cancer Detection"

_applsci, doi:10.3390/app13179982_

Round 1

Reviewer 1 Report

Overall, the manuscript appears to be well-written. In addition, I think that the manuscript might deserve publication in  Applied Sciences after some points are dealt with and some missing details are added prior to publication as follows:

1-      Please discuss in the text the Light-based breast cancer techniques.

2-      Please address in the text the analytical techniques based on nanomaterials for the detection of breast cancer biomarkers.

3-      Please mention in the text the sensitivity of the different breast cancer sensing methods.

4-      In addition to the References list, other current research on the use of laser light as a PT tool is proposed to be reviewed and included if they are beneficial:

Ø    “Early detection of lung cancer biomarkers through biosensor technology: A review”. J Pharm Biomed Anal. 2019 Feb 5;164:93-103.

https://doi: 10.1016/j.jpba.2018.10.017

Ø    “Review of Laser Raman Spectroscopy for Surgical Breast Cancer Detection: Stochastic Backpropagation Neural Networks” Sensors 2020, 20, 6260. https://doi.org/10.3390/s20216260

Ø    “Tunable femtosecond laser suppresses the proliferation of breast cancer in vitro”. J Photochem Photobiol B. 240, (2023) 112665. doi: 10.1016/j.jphotobiol.2023.112665.

Overall, the manuscript appears to be well-written.

Author Response

Here we have included  response letter to reviewer 1.

Reviewer 2 Report

Here are following observations.

1. The manuscript is seems to be more suitable for any book or book chapter rather than a journal paper (even in review mode).

2. The paper is primarily discussing sensing techniques based on microwave frequency and UV range. There are other methodologies also exist like MRI for breast cancer scanning.

3. Section 2 seems to be misleading as Biomarkers as title. However, there is not much details discussion on biomarkers used in the literature for breast cancer detection. 

4. There should be a discussion on such datasets availability for completeness of the paper.

5. Paper title also needs to revise suitably.

6. The review manuscript must have a in depth survey on a particular topic which is lacking at many places in the manuscript and hence leads to poor readability for readers.

Author Response

Here we have included response letter to reviewer 2.

Reviewer 3 Report

1.       The authors prepared a manuscript that discusses the global problem of breast cancer and the limitations of current technology in detecting it early. The authors review microwave sensing and biosensing techniques as potential methods to identify cancer markers at early stages. The reason for conducting this review is the vast number of papers published in this area, highlighting the need for new approaches in breast cancer detection.

The content of this article needs improvement as it is currently unclear and difficult to understand.

2.       Upon reviewing your paper titled "Sensing techniques for breast cancer detection," I noticed that your abstract primarily focuses on microwave techniques. I believe it would be beneficial to modify the title to better align with the content discussed in the paper. A more appropriate title could be "Microwave sensing techniques for breast cancer detection" or "Exploring the potential of microwave sensing for breast cancer detection."

This modification will help accurately reflect the specific area of focus within your research, enabling readers to better gauge the relevance of your paper to their interests.

3.       In the abstract, the authors have made a claim stating that the current technologies utilized for breast cancer detection are inadequate in terms of early detection. However, the specific technologies being referred to are not mentioned.

4.       Some grammatical errors “existing technology or existing technologies” and so on.

5.        Line 20-21 introduction “The behavior of breast cancer is considered for their classification.” the sentence is not scientifically correct. The pronoun "their" is not appropriate when referring to breast cancer, as cancer is not a plural entity. Additionally, the sentence is vague and does not clearly convey the intended meaning.

6.       Paraphrase and check the mean and scientifically soundness of each sentence of your paper.

7.       The authors stated that "Among eight women in the United States, one woman has encountered breast cancer…" I would like to ascertain if this article pertains exclusively to the United States or if it includes global statistics. Please refer to global statics.

8.       Line 48-85 has not any citation or reference.

9.       I would like suggest some paper about electrochemical approaches to use them if you are interested like:

https://doi.org/10.1016/j.biosx.2023.100331

https://doi.org/10.1016/j.talo.2023.100215

https://doi.org/10.1016/j.sbsr.2021.100449  

10.   The conclusion section is better to improve and extend it please

11.   Overall, I would suggest publishing the manuscript as it is well-written and provides important advances in the field of breast cancer. The paper emphasizes the global problem of breast cancer and the significance of early detection for designing individualized treatment plans. It includes a comprehensive review of various methods for the early detection of biomarkers associated with breast cancer, including the newly developed microwave sensing and biosensing techniques. The discussion covers the advantages, limitations, and future research directions of these techniques.

Minor English correction is needed 

Author Response

Here we have included response letter to reviewer 3.

Round 2

Reviewer 1 Report

In response to my previous suggestions and concerns, the authors have made reasonable changes to the manuscript. Overall, the manuscript reads well and clarifies the authors' work. In my opinion, the manuscript contains currently all the information and is ready for publishing in the Journal "Applied Sciences" as a regular article.

Overall, the manuscript reads well and clarifies the authors' work. 

Reviewer 2 Report

Accepted